# Liquid-Phase Adsorption Behavior of β-D-Glucooligosaccharides When Using Activated Carbon for Separation, and the Antioxidant Stress Activity of Purified Fractions

**DOI:** 10.3390/foods13111634

**Published:** 2024-05-24

**Authors:** Guan-Hua Ma, Si-Qi Jiang, Li-Ping Liu, Jie Feng, Jing-Song Zhang, E-Xian Li, Shu-Hong Li, Yan-Fang Liu

**Affiliations:** 1Institute of Edible Fungi, Shanghai Academy of Agricultural Sciences, Key Laboratory of Edible Fungi Resources and Utilization (South), Ministry of Agriculture, National Engineering Research Center of Edible Fungi, Shanghai 201403, China; 2023208016@stu.njau.edu.cn (G.-H.M.); l1228llp@163.com (L.-P.L.); sytufengjie@163.com (J.F.); syja16@saas.sh.cn (J.-S.Z.); 2Biotechnology and Germplasm Resources Institute, Yunnan Academy of Agricultural Sciences, Kunming 650205, China; xiaogaogao4850@126.com (E.-X.L.); shuhongfungi@126.com (S.-H.L.)

**Keywords:** β-glucooligosaccharides, adsorption mechanism, separation, antioxidative stress activity

## Abstract

The adsorption characteristics of β-glucooligosaccharides on activated carbon and the purification were systematically investigated. The maximum adsorption capacity of activated carbon reached 0.419 g/g in the optimal conditions. The adsorption behavior was described to be monolayer, spontaneous, and exothermic based on several models’ fitting results. Five fractions with different degrees of polymerization (DPs) and structures of β-glucooligosaccharides were obtained by gradient ethanol elution. 10E mainly contained disaccharides with dp2a (G1→6G) and dp2b (G1→3G). 20E possessed trisaccharides with dp3a (G1→6G1→3G) and dp3b (G1→3G1→3G). 30E mainly consisted of dp3a and dp4a (G1→3G1→3(G1→6)G), dp4b (G1→6G1→3G1→3G), and dp4c (G1→3G1→3G1→3G). In addition to tetrasaccharides, 40E and 50E also contained pentasaccharides and hexasaccharides with β-(1→3)-linked or β-(1→6)-linked glucose residues. All fractions could inhibit the accumulation of intracellular reactive oxygen species (ROS) in H_2_O_2_-induced Caco-2 cells, and they could improve oxidative stress damage by increasing the activity of superoxide dismutase (SOD) and reduced glutathione (GSH), which were related to their DPs and structures. 50E with high DPs showed better anti-oxidative stress activity.

## 1. Introduction

*Ganoderma*, known as “Lingzhi”, is a member of the fungal family. Its fruit body has been used as a traditional medicine in China for 4000 years [1]. There are a variety of components in *Ganoderma lucidum*, such as polysaccharides, triterpenes, nucleosides, and sterols [2]. *Ganoderma lucidum* contains polysaccharides, which are crucial compounds with diverse biological functions such as immune modulation [3], tumor inhibition [4], liver protection [5], and hypoglycemic effects [6]. In our previous work, a β-glucan component GLP20 was purified from the fruit body of *Ganoderma lucidum*. The β-glucan exhibited favorable biological activity [7] with a molecular weight of 2.42 × 10^6^ g/mol. It consisted of β-(1→3)-glucose as the primary chain and β-(1→6)-glucose as the branched chain connecting glucan. The ratio between the main chain and the branched chain was 3:1 [7]. However, the extracted polysaccharides exhibited high molecular weight, poor bioavailability, and low solubility, which greatly limited its development and effective utilization [8]. According to a previous report, the degradation of polysaccharides to obtain low-molecular-weight oligosaccharides not only preserves the diverse biological functions of polysaccharides but also offers an effective solution to the aforementioned issues. Presently, studies concerning the degradation of β-(1→3)-glucans without branches and the corresponding activity of glucooligosaccharides have been reported [9,10]. However, the literature on the degradation of β-(1→3)-glucans with branches at the O-6 position, especially for *Ganoderma lucidum* β-glucan, is scarce. Moreover, the mechanism of the relevant activity and the structure–activity relationship of the corresponding glucooligosaccharides have not yet been clarified. In our previous study, GLPW, a mixture of glucooligosaccharides obtained by microwave degradation of GLP20, was proven to have good anti-inflammatory activity [11]. GLPW was preliminarily separated using the ethanol precipitation method to acquire four fractions varying in polymerization degree. The pharmacological activity findings indicated that the fractions with a polymerization degree of 2–8 (GLPWA) showed better anti-inflammatory activity [12]. However, since the glucooligosaccharides degraded by glucan contained fragments with different degrees of polymerization (DPs), and the mixed products were usually used for activity testing, the main active fragments in the products were unclear. Hence, to examine the impacts of DPs and the structural properties of glucooligosaccharides on functionalities, further separation and purification are necessary.

In recent times, numerous techniques have been devised for the implementation of oligosaccharide separation and purification [13]. Anion-exchange chromatography [14], gel exclusion chromatography [15,16], and high-performance liquid chromatography [17] are the three most commonly employed techniques for effectively separating oligosaccharides. However, these separation methods suffer from the disadvantage of being both time-consuming and inefficient. Activated carbon, known for its porous structure and low cost, is the most widely used and most effective adsorbent. Its surface was found to contain numerous functional groups and, therefore, had exceptional adsorption ability for various inorganic and organic substances in gas and liquid phases [18,19]. Diverse organic molecules could bind to it through various physical and chemical mechanisms and forces, including Van der Waals, hydrogen bonding, etc. [20]. Activated carbon has wide application prospects in the process of oligosaccharides’ separation and purification [21]. The use of ethanol aqueous solution at varying concentrations resulted in different elution selectivity for the recovery of oligosaccharides during activated carbon treatment. Commercial galactooligosaccharide mixtures were eluted by activated carbon column chromatography with 10% ethanol, which mainly yielded galactooligosaccharides (DP3–8) with good separation [22]. It should be noted that in these applications, the loss of oligosaccharide samples was often serious due to the lack of theoretical guidance. The primary cause was rooted in the adsorption interplay between oligosaccharides and activated carbon. Meanwhile, relatively few studies have been conducted on activated carbon for the separation of β-glucooligosaccharides. To our knowledge, there have been no reports on the optimal conditions for adsorption and the mechanism of β-glucooligosaccharides on activated carbon, which limits the application of activated carbon in β-glucooligosaccharides’ processing to a certain extent.

In this study, the adsorption characteristics of β-glucooligosaccharides (GLPWA, DP 2~8) on activated carbon were systematically studied with different experimental conditions, including the effects of the activated carbon addition ratio, solution pH, adsorption temperature, and time on the adsorption behavior. The kinetic and thermodynamic mechanisms were further explored. In addition, β-glucooligosaccharides with narrower DPs were isolated by desorption with different concentrations of ethanol, and the antioxidative stress activities of the isolated products were compared and evaluated. The objective of this research was to examine the process of β-glucooligosaccharides’ adsorption on activated carbon and analyze the impact of β-glucooligosaccharides’ degree of polymerization on their ability to mitigate oxidative stress damage, thus offering assistance for future studies and efforts aimed at the utilization of β-glucooligosaccharides.

## 2. Materials and Methods

### 2.1. Chemicals and Reagents

*Ganoderma lucidum* β-glucooligosaccharides (GLPW) (DP2–24, purity > 90.11%) were prepared by microwave degradation from β-glucan GLP20 containing β-(1→3)-linked D-glucan with β-(1→6)-D-glucose branches [12]. β-glucooligosaccharides (GLPWA) (DP2~8, purity > 92.70%) were prepared by 90% ethanol precipitation from GLPW [12]. Activated carbon was sourced from Sigma-Aldrich (St. Louis, MO, USA). The Caco-2 cell was derived from the Type Culture Collection of the Chinese Academy of Sciences (Shanghai, China). DMEM basic (1×) was obtained from Gibco (Grand Island, NE, USA). Hydrogen peroxide (H_2_O_2_) solution (30%, *w*/*w*) used as an inducer of oxidative stress was sourced from Sigma-Aldrich (St. Louis, MO, USA). The total protein assay kit, superoxide dismutase (SOD), and reduced glutathione (GSH) assay kit were bought from Nanjing Jiancheng Co., Ltd. (Nanjing, China). The RIPA lysis buffer, phenylmethylsulfonyl fluoride (PMSF), and reactive oxygen species (ROS) assay kit were bought from Beyotime Biotechnology (Shanghai, China). 

### 2.2. Batch-Mode Adsorption Studies

The adsorption of β-glucooligosaccharides on activated carbon was investigated through batch adsorption equilibrium experiments (as illustrated in Figure 1).

β-glucooligosaccharides (GLPWA) were dissolved in water at the concentration of 10 g/L, and different proportions of activated carbon (β-glucooligosaccharides: activated carbon = 1:2, 1:3, 1:4, g/g) were added. The centrifuge tube was placed in an oscillating instrument with a constant temperature of 25 °C and shook for a duration of 12 h. 

To investigate the impact of pH on adsorption, the pH of the β-glucooligosaccharide solution was modified to a range of 2 to 11 by introducing diluted solutions of HCl or NaOH.

In order to study the impact of temperature on adsorption, experiments were carried out at different temperatures including 25, 35, 45, 55, and 65 °C, respectively. Additionally, the adsorption of β-glucooligosaccharides was tested at various time intervals (0–540 min) to determine the effect of contact time.

To observe the effect of the initial concentration of β-glucooligosaccharides on adsorption and to fit the adsorption isotherms, parallel experiments were carried out at 25, 35, 45, and 55 °C, and the initial concentrations of β-glucooligosaccharides were 1.0, 2.0, 4.0, and 6.0 g/L, respectively.

After adsorption, the supernatant solution was filtered, and the composition of β-glucooligosaccharides in the filtrate was analyzed via high-performance anion exchange chromatography [23]. The concentration of β-glucooligosaccharides in the residues was measured by the phenol-sulfuric acid method [24], and the rate of adsorption on activated carbon was calculated. The formula is as follows:n (%) = (C_0_ − C_t_)/C_0_ × 100(1)

Based on the change in solution concentration before and after adsorption, the adsorption amount of β-glucooligosaccharides was calculated as follows:q = (C_0_ − C_t_)/m × V(2)
where C_0_ and C_t_ are the initial and final concentrations of β-glucooligosaccharides, respectively. V and m are the volume of the solution and the addition amount of activated carbon, respectively.

### 2.3. Theoretical Study of the Adsorption of β-Glucooligosaccharides by Activated Carbon

#### 2.3.1. Adsorption Kinetics Analysis

Batch adsorption tests were carried out on the activated carbon adsorption of β-glucooligosaccharides at different time intervals to determine the process involved. The adsorption data obtained at different times were examined using different kinetic models called pseudo-first-order (PFO) [25] and pseudo-second-order (PSO) [26].

The pseudo-first-order Lagergren model of the solid/liquid adsorption system is expressed as
q_t_ = q_1_ × (1 − e^−k1t^)(3)
where q_1_ and q_t_ (g/g) are the amounts of β-glucooligosaccharides adsorbed at equilibrium and at any time t, respectively. k_1_ represents the rate constant of adsorption, which follows a pseudo-first-order reaction (min^−1^).

The pseudo-second-order kinetic model assumes that the adsorption rate is proportional to the square of the number of unoccupied sites. The model can be expressed as follows:q_t_ = k_2_q_2_^2^t/(1 + k_2_q_2_t)(4)
where k_2_ is the pseudo-second-order adsorption rate constant (g/mg/min). The pseudo-second-order rate constants k_2_ and q_2_ are calculated by using the slope and intercept of t/q_t_ for t. 

#### 2.3.2. Adsorption Equilibrium Isotherm

Studies have shown that there is a relationship between the adsorption capacity and adsorbate concentration. In order to further understand the mechanism for the adsorption process of activated charcoal, two different models were applied to analyze this process, the Freundlich and the Langmuir equations. 

The Langmuir model is expressed as
q_e_ = q_max_K_L_C_e_/(1 + K_L_C_e_)(5)
where q_max_ (g/g) and K_L_ (L/g) are the Langmuir constants related to the adsorption capacity and adsorption energy, respectively.

The Freundlich model is expressed as
q_e_ = K_f_C_e_^n^(6)
where C_e_ and q_e_ are the concentration (g/L) and the adsorption value (g/g) of β-glucooligosaccharides in the adsorption equilibrium state, respectively. K_f_ (L/g) and n are isotherm constants and the exponent for Freundlich.

#### 2.3.3. Thermodynamic Study

To investigate the spontaneous adsorption of β-glucooligosaccharides on activated carbon, experiments were conducted at various temperatures. Numerous thermodynamics parameters were found using the below equations.

The thermodynamic parameters were further calculated from the Langmuir equilibrium constant (K_L_). The thermodynamic equation is as follows [27]:ln K_L_ = −∆H^0^/RT + (∆G^0^)/R(7)
and
∆G^0^ = ∆H^0^ − T∆S^0^(8)
where R is the universal gas constant (8.314 J mol^−1^ K^−1^) and T is the absolute temperature (in Kelvin).

### 2.4. Desorption Separation of GLPWA

Activated carbon that had reached adsorption equilibrium was collected, the unadsorbed β-glucooligosaccharides were removed by washing with deionized water, and then the carbon was eluted using a gradient of ethanol to obtain several fractions. Firstly, 10% (*v*/*v*) ethanol was added with stirring for 60 min and filtered out. The elution procedure was repeated three times to collect the filtrate for concentration, and this was freeze-dried to obtain fraction 10E. Then, 20% (*v*/*v*) was added to the activated carbon obtained by filtration, and the above step was repeated to obtain fraction 20E. Subsequently, 30%, 40%, and 50% (*v*/*v*) ethanol solutions were added sequentially to obtain the fractions (named 30E, 40E, 50E) (as shown in Figure 1). 

### 2.5. Sugar Content, Polymerization Analysis, and Structural Identification of Glucooligosaccharides

The dilution factor of the sample should be determined first. When diluted to the appropriate concentration, 1 mL of the sample solution should be mixed with 0.5 mL of phenol by vortexing. Then, 2.5 mL of concentrated sulfuric acid should be added and mixed by vortexing. The resulting mixture should be placed in a water bath at 100 °C for 15 min to allow for reaction. After cooling with cold water, 200 μL of the solution should be transferred onto an enzyme plate and the absorbance value at 490 nm should be measured.

Polymerization characteristic analysis of components in the fractions was performed by high-performance anion exchange chromatography (HPAEC) equipped with a pulsed amperometric detector (PAD). Samples were dissolved in deionized water (25 μL) prior to injection and separated in the chromatograph using a linear gradient elution method with a CarboPac PA-100 column (4 × 250 mm) and a CarboPac PA-100 guard column (3 × 50 mm). The operation was conducted at a column temperature of 30 °C and a flow rate of 1 mL/min. The mobile phase consisted of eluent A (150 mM NaOH) and eluent B (100 mM NaOH containing 500 mM CH_3_COONa). The elution gradient for analysis was as follows: the percentage of eluent A changed from 90% at 0 min to 30% at 30 min, and the percentage of eluent B changed from 10% at 0 min to 70% at 30 min. Subsequently, a 100% eluent B wash was performed on the column for 10 min, followed by re-equilibration with a mixture of 90% eluent A and 10% eluent B for 5 min after each run.

### 2.6. Assays of Antioxidative Stress Activity In Vitro

#### 2.6.1. Establishment of Cell Model In Vitro and Protective Experiments’ Design

Caco-2 cells were grown in DMEM containing 10% heat-treated FBS and 1% penicillin-streptomycin at a temperature of 37 °C with 5% CO_2_. To induce oxidative stress, Caco-2 cells were treated with various concentrations (100, 200, 300, 400, 600, 800, 1000, 1500, 2000 μmol/L) for different hours (6 h or 12 h) to gain the desired concentration and time to establish cell model. In protective experiments, the cells were pretreated by different fractions at 10, 50, and 200 µg/mL for 12 h, followed by treatment with H_2_O_2_ for 6 h. Then, the corresponding tests were performed.

#### 2.6.2. Cell Viability Assay

The cells were placed in 96-well trays containing 200 µL of growth medium and incubated at 37 °C in a humidified incubator with 5% CO_2_. After being treated with H_2_O_2_ or β-glucooligosaccharides, 30 µL of Alamar Blue with a concentration of 0.1 mg/mL was introduced into every well. The plate was then placed in the incubator for further incubation until the blank control group exhibited alterations in color. The measurement of absorbance was conducted at wavelengths of 570 nm and 600 nm, and the cell survival rate was calculated according to the formula [28]. 

#### 2.6.3. Detection of Intracellular Reactive Oxygen Species

Before staining with 2,7-dichloride-hydrofluorescein diacetate (DCFH-DA), cells were placed in test tubes and washed twice with PBS. The combination of ROS and DCFH-DA resulted in the production of dichlorofluorescein (DCF), a compound that emits green fluorescence. The level of ROS was determined by measuring the mean fluorescence intensity using a flow cytometer. In another experiment, adherent cells were washed with PBS and then incubated with DCFH-DA before being photographed and analyzed under a multifunctional enzyme marker.

#### 2.6.4. Determination of SOD and GSH In Vitro

After being treated with β-glucooligosaccharides and H_2_O_2_, RIPA and PMSF were used to lyse the cells. The lysate was centrifuged at 12,000× *g* and 4 °C for 20 min and the supernatant was collected; then, the total protein assay kit was used to determine protein concentrations. The SOD enzymatic activity was measured by a superoxide dismutase assay kit, and the GSH intracellular content was detected using a reduced glutathione assay kit. The levels of SOD and GSH were determined using kits following the manufacturer’s instructions.

### 2.7. Statistics

The data were presented as the mean ± standard deviation. One-way analysis of variance (ANOVA) was used to analyze differences between groups at significance levels of *p* < 0.05 and *p* < 0.01.

## 3. Results

### 3.1. Effect of Adsorption Conditions on Adsorption

#### 3.1.1. Effect of Activated Carbon Addition Ratio on Activated Carbon Adsorption of GLPWA

In order to achieve complete adsorption of β-glucooligosaccharides, the optimal ratio of GLPWA to added activated carbon amounts was investigated. The results (Figure 2A) showed that the adsorption rate increased with the increase in activated carbon addition. When the addition ratio of β-glucooligosaccharides to activated carbon was 1:2, the adsorption rate was only 53.82%. Combined with the composition analysis performed by HPAEC (Figure 2B), it was found that β-glucooligosaccharides with larger DPs in a retention time of 15–30 min were left in the solution, and only low DPs were adsorbed. When the ratio was changed to 1:3, a small amount of β-glucooligosaccharides with larger DPs remained in the sample solution. When the addition ratio reached 1:4, the adsorption rate reached 86.39% and only glucose was left in the solution, indicating that the rest of the components were almost completely adsorbed.

From these phenomena, we elucidated that DPs had effects on the adsorption of activated carbon, and the components with smaller DPs were preferentially adsorbed. The interaction strength between β-glucooligosaccharides and activated carbon increased with the increase in the molecular weight of sugar [29,30]. Therefore, it was shown that β-glucooligosaccharides were adsorbed more than glucose. To achieve complete adsorption, the ratio of 1:4 was chosen for the addition of activated carbon.

#### 3.1.2. Effect of pH on Activated Carbon Adsorption of GLPWA

The pH value is a crucial factor affecting activated carbon’s adsorption performance. Hydrogen and hydroxide ions impact the adsorption process by dissociating the groups on the adsorbent and adsorbate [31]. In this research, the adsorption capacity of β-glucooligosaccharides on activated carbon at pH 2–12 was studied. As shown in Figure 2C, in the range of pH 2–3, the adsorption was low. With the increase in pH, the adsorption increased gradually, and it reached the highest at pH 5, with a value of 0.467 g/g. Nevertheless, as the pH level rose, the adsorption declined within the pH range of 7–9; however, with a further increase in pH, the adsorption experienced a slight increase. The maximum adsorption occurred in the nearly neutral solution with a pH of 5–6, indicating that an excess of H^+^ or OH^-^ was not conducive to the adsorption of β-glucooligosaccharides on activated carbon. Oligosaccharides may be partially hydrolyzed under a strong acidic condition, which results in a decrease in adsorption. The results indicated that activated carbon adsorbed β-glucooligosaccharides more efficiently under slightly acidic conditions. pH 5 was used for the following batch-mode experiments. 

#### 3.1.3. Effect of Temperature on Activated Carbon Adsorption of GLPWA

The adsorption behavior of activated carbon might be affected by temperature, depending on the properties of adsorbents and adsorbed substances [32]. As shown in Figure 2D, there was no significant difference in the adsorption capacity of activated carbon with the increase in temperature in the range of 25–65 °C. The results showed that the adsorption capacity of activated carbon on β-glucooligosaccharides was slightly affected by temperature. The adsorption capacity of activated carbon for β-glucooligosaccharides was found to be insensitive to temperature, in contrast to inulin oligosaccharides, which exhibited a decrease in adsorption capacity with increasing temperature [33]. This difference might be due to the different structures and functional groups of the two kinds of oligosaccharides. Therefore, the experiment was carried out at room temperature (25 °C).

#### 3.1.4. Effect of Time on Activated Carbon Adsorption of GLPWA

The adsorption capacity of GLPWA by activated charcoal was investigated while varying the shaking time from 0 to 540 min in the solution of pH 5 at 25 °C. As shown in Figure 2E, the adsorption of GLPWA on activated charcoal mainly occurred in two distinct stages. The adsorption amount of β-glucooligosaccharides reached 0.365 g/g at 5 min. After 5 min, the adsorption capacity increased slowly, and it reached the equilibrium after 60 min, with the maximum adsorption capacity of 0.419 g/g. The majority of β-glucooligosaccharides were adsorbed within the first 5 min, with approximately 73% of GLPWA being adsorbed. This adsorption phenomenon was similar to the behavior of activated carbon adsorption of chitooligosaccharides [34]. These two phases corresponded to the phenomena of surface adsorption and intra-particle diffusion, respectively [32].

In brief, the maximum adsorption capacity could be up to 0.419 g/g when the activated carbon addition ratio is 1:4 and it is adsorbed at pH 5 and 25 °C for 60 min.

### 3.2. Theoretical Study on the Adsorption of β-Glucooligosaccharides by Activated Carbon

#### 3.2.1. Adsorption Kinetics Analysis

The information collected on the impact of the contact duration was utilized to establish the sorption kinetics of β-glucooligosaccharides. By fitting the experimental data of the adsorption time with pseudo-first- and pseudo-second-order models, valuable information was obtained on the properties of the adsorption process, thus revealing its kinetic mechanism.

The kinetic model fitting curves are presented in Figure 3, and the parameters are shown in Table 1. It can be seen from Table 1 that the R^2^ of the pseudo-first- and pseudo-second-order kinetic models were 0.9729 and 0.9871, respectively, and the pseudo-second-order model could better describe the β-glucooligosaccharide adsorption mechanism. Furthermore, the q_2_ value determined using the pseudo-second-order kinetic model exhibited a higher proximity to the experimental q_e_ value (0.419 g/g). The findings showed that the adsorption mechanism of the β-glucooligosaccharides on activated carbon followed a pseudo-second-order kinetic reaction.

#### 3.2.2. Adsorption Isotherms’ Analysis

Understanding the trend and degree of the adsorption process through the study of adsorption thermodynamics is crucial for explaining the adsorption mechanism and law. The relationship between the concentration of the residual adsorbate in the solution and the amount of adsorbate at equilibrium is represented by the adsorption isotherm. The adsorption equilibrium refers to the state when the concentration of the adsorbate in the solution and the concentration on the surface of the adsorbent do not change [35]. 

Two models—the Freundlich equation and the Langmuir equation—were used for the analysis, to further our understanding of the adsorption mechanism of activated carbon. The Langmuir isotherm assumes a monolayer adsorption process with a uniform, finite number of energetic adsorption sites, whereas the Freundlich isotherm posits monolayer sorption occurring at active sites with varying levels of energy distribution [36,37].

The fitting curves and correlation parameters are exhibited in Figure 3D,E and Table 2. The R^2^ values indicated that the Langmuir model could better fit the adsorption experimental data, revealing that the adsorption process mainly involved monolayer adsorption with identical active sites.

#### 3.2.3. Thermodynamic Studies

The values of ∆G^0^, ∆H^0^, and ∆S^0^ are shown in Table 2. The β-glucooligosaccharide adsorption on activated carbon was shown by the negative ∆G^0^ values to be spontaneous. The decrease in the calculated value of ∆G^0^ with increasing temperature suggested a decrease in the spontaneity of the reaction. However, an increase in temperature within a certain range could also promote adsorption. Higher temperatures resulted in greater disorder and potentially enhanced effective contact for improved adsorption. The negative value of ∆H^0^ (−5.220 kJ/mol) indicated that the adsorption reaction of β-glucooligosaccharides is exothermic. Glucooligosaccharides contain plenty of hydroxyl groups, which could form hydrogen bond interactions with the molecules on the surface of activated charcoal and in that way be absorbed. The adsorption process occurred rapidly due to the low adsorption heat between the activated carbon surface and the β-glucooligosaccharide molecules. In addition, the results showed that the entropy of β-glucooligosaccharides decreased during the adsorption process on activated carbon, which might have been since the movement of molecules was limited when β-glucooligosaccharides were adsorbed on the surface of activated carbon, resulting in a decrease in entropy [33].

### 3.3. Desorption Separation and Characterization Analysis of GLPWA

As in the previous adsorption study (in Figure 2B), sugars were selectively adsorbed onto the activated carbon, with β-glucooligosaccharides more strongly adsorbed than monosaccharides. Ethanol gradient elution was a frequently employed technique for separating oligosaccharides through activated carbon chromatography [38,39]. Some studies used water or very little ethanol to separate monosaccharides, 5–10% ethanol aqueous solution to recover disaccharides, and 15–50% ethanol aqueous solution to recover oligosaccharides [21,40,41]. Therefore, it was proposed that the separation of β-glucooligosaccharides with different DPs would be achieved using gradient ethanol elution. After being desorbed, six fractions were obtained from GLPWA based on different concentrations (0–50%, *v*/*v*) of ethanol elution. The efficient recovery and total sugar contents of fractions eluted by different percentages of ethanol are shown in Table 3. The total recovery was 84.53%, and the total sugar content was higher than 75% in all cases except for 20E. 

In a previous study, the structures of oligosaccharides with DP (2–6) in GLPWA were identified as linear β-(1→3)-linked glucooligosaccharides or ones with β-(1→6)-linked glucose residues based on HPAEC-MS/MS analysis [12]. The main components in each fraction after separation could be identified accordingly. The main component of pure water elution was glucose (Figure 4A). As shown in Figure 4A,C, the 10E fraction obtained by 10% ethanol elution was mainly composed of β-glucooligosaccharides of DP2, with structures of dp2a (G1→6G) and dp2b (G1→3G). The 20E fraction eluted with 20% ethanol mainly contained β-glucooligosaccharides of DP3, with structures of dp3a (G1→6G1→3G) and dp3b (G1→3G1→3G). The 30E fraction eluted with 30% ethanol was mainly rich in β-glucooligosaccharides of DP3 and DP4, with structures of dp3a and dp4a (G1→3G1→3(G1→6)G), dp4b (G1→6G1→3G1→3G), and dp4c (G1→3G1→3G1→3G). Most of the major peaks in GLPWA could be eluted with 10–30% ethanol. Magnifications of HPAEC analysis spectra for 40E and 50E fractions eluted by 40% and 50% ethanol (Figure 4B) showed that higher concentrations of ethanol could elute β-glucooligosaccharides with higher DPs. For 40E, except for dp4a, dp4b and dp4c, three pentasaccharides (dp5a (G1→6G1→3G1→3G1→3G), dp5b (G1→3G1→3(G1→6)G1→3G), and dp5c (G1→3G1→3G1→3G1→3G)) and a hexasaccharide (dp6a (G1→6G1→3G1→3G1→3G1→3G)) were detected. 50E mainly contained fractions with various glucooligosaccharide isomers of DP4–DP8.

Overall, according to the desorption capacity of the glucooligosaccharides in the ethanol solution, 10–50% ethanol successfully separated GLPWA into five fragments containing β-glucooligosaccharides with different DPs. Considering that the recovery rate of the method was high, the technique was straightforward and efficient in the isolation, enabling the gathering of five β-glucooligosaccharide segments for analyses of the functionality and the correlation between bioactivity and DPs.

### 3.4. Construction of Oxidative Stress Model and Activity Evaluation

In a previous study, the anti-inflammatory activity of four fractions with different DPs obtained by graded alcohol precipitation was investigated, and the results showed that the fractions with lower DPs (GLPWA, DP2–8) exhibited superior anti-inflammatory activity [12]. In order to further investigate the relationship between DPs and function, an oxidative damage model was used in this study. Numerous studies have indicated that oxidative damage leads to a defect in barrier function, which is associated with the occurrence of intestinal diseases [42]. Therefore, controlling oxidative damage offers an effective way to prevent a variety of intestinal diseases.

#### 3.4.1. Oxidative Stress Model and Cell Damage Repair Ability

Exogenous H_2_O_2_ is often used as a representative inducer of the oxidative stress model [43]. According to the research, oxidative stress can result from either excessive production of ROS or a decrease in antioxidants, both of which were linked to cell demise [44,45]. Experiments were carried out with cell viability as an indicator to determine the appropriate concentration of H_2_O_2_ [46,47].

The results (Figure 5A) showed that the cell viability of Caco-2 cells decreased in a dose-dependent manner with the increase in H_2_O_2_ concentration. When the H_2_O_2_ concentration was less than 200 μmol/L, the survival rates of cells incubated for 6 or 12 h were still above 60%. When the concentration of H_2_O_2_ was 400 μmol/L and the injury time was 6 h, the cell survival rate was 53%. Consistent with prior research [48], it was observed that a substantial amount of H_2_O_2_ hindered cell proliferation or exhibited cytotoxic effects on the cells. Therefore, 400 μmol/L H_2_O_2_ was chosen to damage cells for 6 h to establish the oxidative stress injury model.

After being treated with various concentrations of β-glucooligosaccharide fractions, the viability of Caco-2 cells showed a significant improvement (Figure 5B). GLPWA and 30E demonstrated excellent protective effects on Caco-2 cells within the concentration range of 10–200 μg/mL. Meanwhile, at a low concentration of 10 μg/mL, GLPWA and 10E showed better protection activity against injury of Caco-2 cells induced by H_2_O_2_.

#### 3.4.2. Measurement of Cellular Levels of ROS, SOD, and GSH

The pathogeneses of numerous diseases are significantly influenced by ROS-induced oxidative harm, which encompasses harm to lipids, proteins, and DNA [49]. In order to examine the safeguarding impact of β-glucooligosaccharides on Caco-2 cells against oxidative stress induced by H_2_O_2_, ROS production was detected using the DCFH-DA probe. As shown in Figure 5C, the fluorescence intensity in the model group was significantly higher than that in the negative control group, indicating that H_2_O_2_ induced cellular oxidative stress by producing excess ROS. Compared with the model group, the fluorescence intensity of the GLPWA group was significantly lower, and it was concentration-dependent. To investigate the protective mechanism of β-glucooligosaccharides against H_2_O_2_-induced oxidative stress damage in Caco-2 cells, a flow cytometer was applied to detect H_2_O_2_-induced ROS. As shown in Figure 5D, Caco-2 cells exposed to 400 μmol/L H_2_O_2_ for 6 h showed a significant increase in intracellular ROS compared to the control group, as evidenced by the shift of the peak to the right. However, β-glucooligosaccharide pretreatment reversed H_2_O_2_-induced oxidative stress damage (Figure 5D,E). The ROS proportion could be significantly reduced by fractions with different DPs. This intervention effectively mitigated the damage caused by oxidative stress and protected the cells. Furthermore, the results suggested a correlation between the DPs of β-glucooligosaccharides and their beneficial effects, wherein pretreatment of cells with 50 μg/mL 20E for 12 h significantly decreased the H_2_O_2_-induced ROS proportion from 100% (H_2_O_2_) to 40.70%, indicating that structures with dp3a (G1→6G1→3G) and dp3b (G1→3G1→3G) exhibit better antioxidant stress activity. Moreover, the ROS content of the 40E and 50E groups was significantly reduced, indicating that glucooligosaccharides with higher DPs and abundant fragments also have a strong ability to reduce oxidative stress damage.

SOD belongs to the antioxidant enzyme system, which has the activity of reactive oxygen scavengers [50]. As shown in Figure 5F, the activity of SOD decreased after H_2_O_2_ damage, and the SOD activity of the model group was 5.16 U/mgprot, which was significantly lower than that of the control group (*p* < 0.01). The SOD activity increased after fractions’ treatment, and the 50E group had a better capacity for increasing the SOD activity of the damaged cells in this experimental concentration range.

GSH is the most important component of the human non-enzymatic antioxidant system, which protects proteins from oxidation and reduces ROS-mediated oxidative damage through reversible post-translational modifications [51]. As shown in Figure 5G, H_2_O_2_ treatment reduced the level of GSH; meanwhile, β-glucooligosaccharide treatment increased it. It was also evident that the β-glucooligosaccharides isolated with activated carbon had a stronger ability to increase GSH activity compared to the GLPWA group. Within the concentration range of the experiment, 50E had the most remarkable effect. However, some high-DP components in the mixtures might play a more important role.

Based on the ROS, SOD, and GSH level analysis results, the 50E fraction exhibited the best antioxidant stress activity among all fractions; considering their structural information, this might have been due to its higher degrees of polymerization (DP4–8) and abundant β-glucooligosaccharide fragments.

## 4. Conclusions

In this study, the optimal adsorption parameters for β-glucooligosaccharide adsorption onto activated carbon were ascertained by batch adsorption experiments. Then, the adsorption mechanism of β-glucooligosaccharides adsorbed on activated carbon was explored. The adsorption reaction was found to follow the pseudo-second-order kinetic model and the Langmuir isotherm, indicating that the adsorption was monolayer and that the adsorption rate decreased with the increase in the initial concentration of β-glucooligosaccharides. Thermodynamic data indicated that the adsorption process was spontaneous and exothermic. Furthermore, GLPWA were separated using gradient ethanol elution to obtain five fractions, 10E, 20E, 30E, 40E, and 50E, which mainly consisted of β-glucooligosaccharides with β-(1→3)-linked or with β-(1→6)-linked glucose residues for different DPs. To further evaluate the protective role of β-glucooligosaccharides in oxidative stress damage, the effect and underlying mechanism of β-glucooligosaccharides acting on the Caco-2 human intestinal epithelial cells was investigated under oxidative stress induced by H_2_O_2_. Under H_2_O_2_-induced oxidative stress, β-glucooligosaccharide pretreatment enhanced Caco-2 cell viability, reduced the activity levels of ROS, and increased the levels of SOD and GSH. The activity was related to the polymerization degree and structure of β-glucooligosaccharides, and the results showed that the 50E component, given its high degree of polymerization and abundant glucooligosaccharide structures, has the best antioxidant stress activity. These findings are of particular importance for the separation and purification of other oligosaccharides using activated carbon, and they also contribute significantly to the investigation of the structure–activity relationship of β-glucooligosaccharides.

## Figures and Tables

**Figure 1 foods-13-01634-f001:**
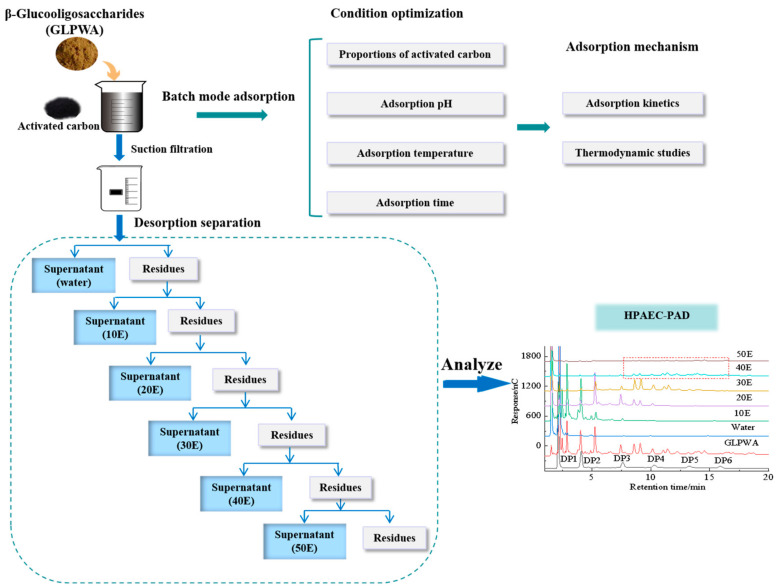
Optimization of conditions for activated carbon adsorption of GLPWA (DP2–8) and gradient ethanol desorption separation experiments.

**Figure 2 foods-13-01634-f002:**
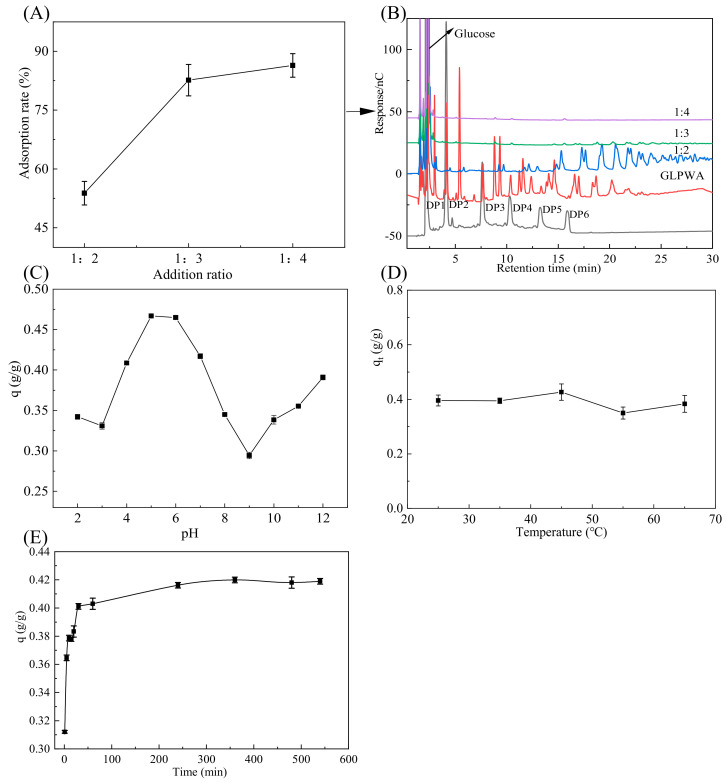
Effect of different adsorption conditions on adsorption. (**A**) Adsorption rates of β-glucooligosaccharides under different ratios of activated carbon addition. (**B**) HPAEC analysis of GLPWA and fractions before and after adsorption on activated carbon with different additive ratios. (**C**) Effect of initial pH on activated carbon adsorption of GLPWA. (**D**) Effect of different temperatures on activated carbon adsorption of GLPWA. (**E**) Effect of adsorption time on activated carbon adsorption of GLPWA.

**Figure 3 foods-13-01634-f003:**
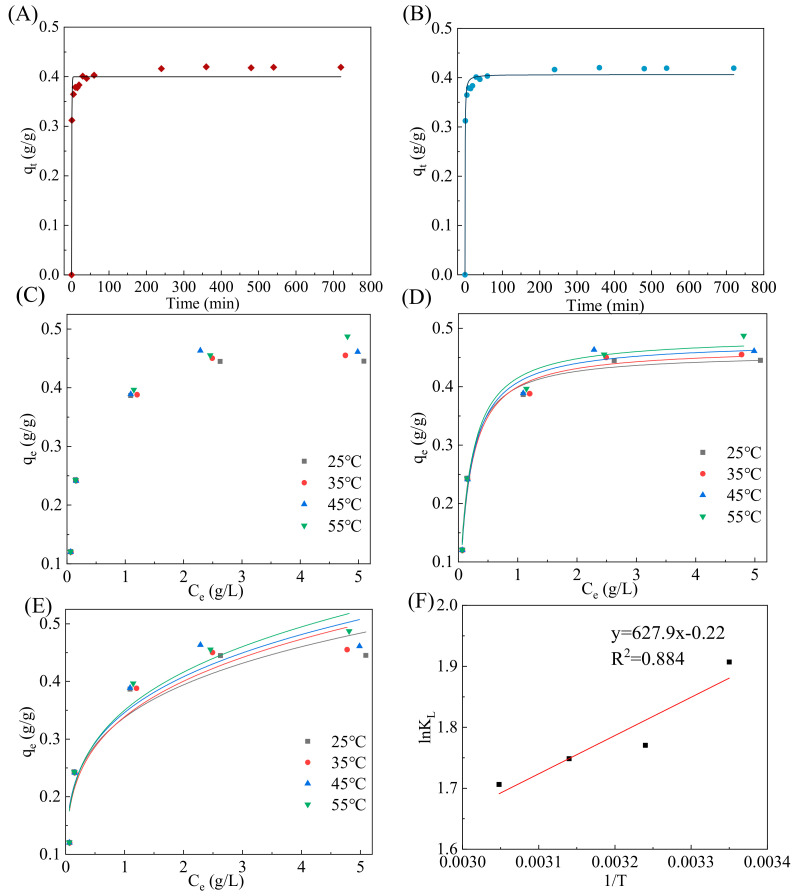
Adsorption kinetics and adsorption isotherms of GLPWA by activated carbon. (**A**) Pseudo-first-order kinetic plots. (**B**) Pseudo-second order kinetic plots. (**C**) Adsorption isotherms. (**D**) Langmuir model. (**E**) Freundlich model. (**F**) Linear correlations between lnk_L_ and 1/T.

**Figure 4 foods-13-01634-f004:**
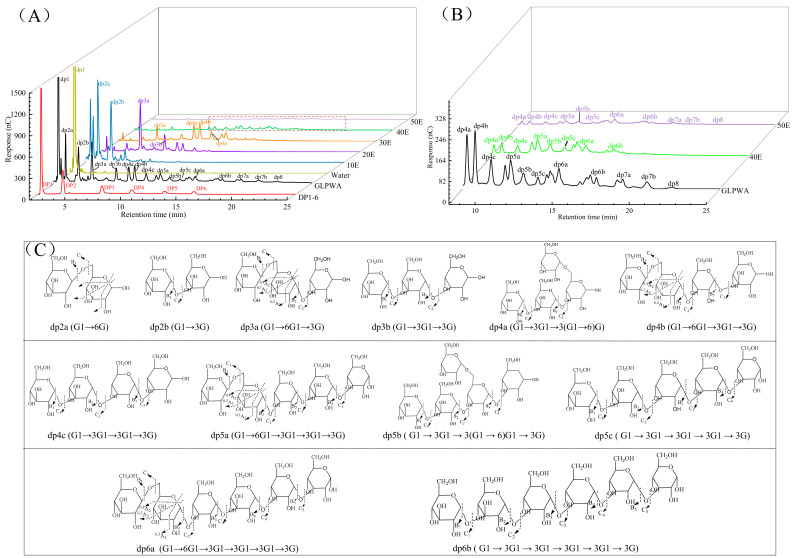
HPAEC analysis of β-glucooligosaccharide in fractions separated by activated charcoal. (**A**) HPAEC elution profiles of GLPWA and separated fractions. (**B**) Local magnification of HPAEC elution profiles for 40E and 50E. (**C**) Structural formulas of β-glucooligosaccharides with different DPs [12].

**Figure 5 foods-13-01634-f005:**
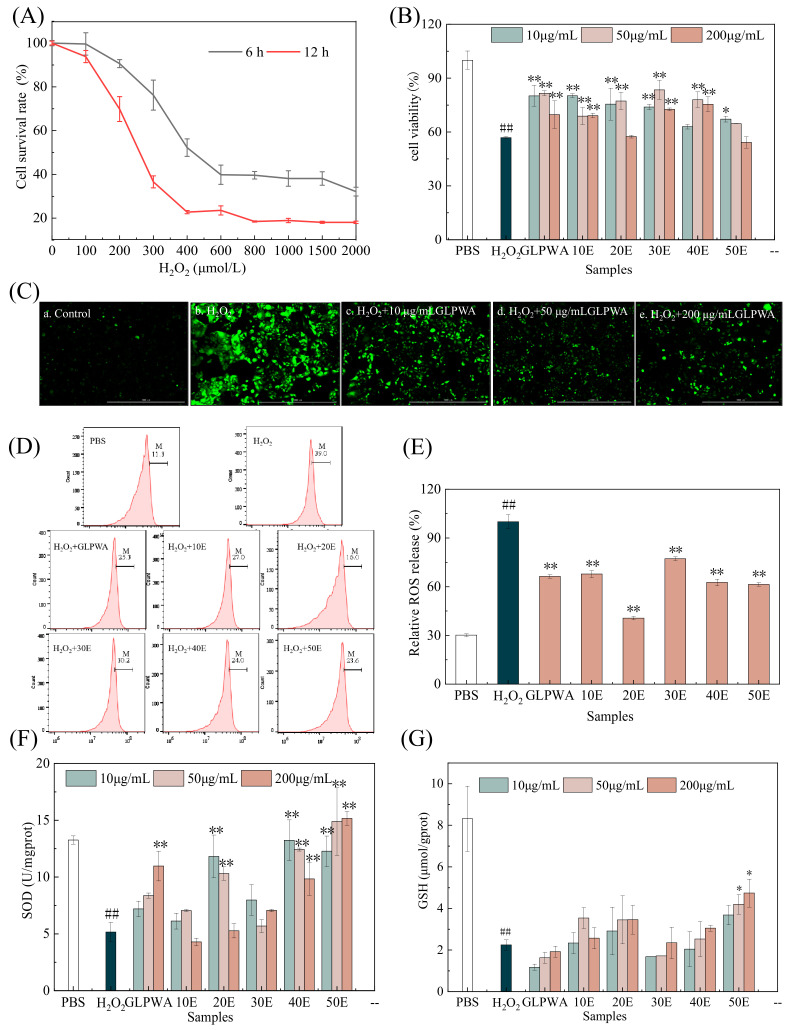
(**A**) The impact of varying concentrations of H_2_O_2_ and durations of induction on the survival rate of Caco-2 cells. (**B**) Effect of GLPWA and its isolated products on the survival rate of Caco-2 cells injured by oxidative stress. (**C**) The ROS fluorescence intensity of model cells under different mass concentrations of GLPWA. (**D**) Results of fluorescence intensity in different groups detected through BD Accuri C6 flow cytometry. (**E**) Relative ROS release (%). (**F**) Effect of GLPWA and its separated components on SOD of Caco-2 cells injured by H_2_O_2_. (**G**) Effect of GLPWA and its separated components on GSH of Caco-2 cells injured by H_2_O_2_. Data are expressed as a percentage of the model group. * *p* < 0.05 vs. H_2_O_2_ group, ** *p* < 0.01 vs. H_2_O_2_ group, ^##^
*p* < 0.01 vs. control group.

**Table 1 foods-13-01634-t001:** Kinetic parameters for GLPWA’s adsorption on activated carbon.

Sample	Pseudo-First-Order Model			Pseudo-Second-Order Model		
β-Glucooligosaccharides	k_1_ (min)	q_1_ (g/g)	R^2^	k_2_ (g g^−1^ min^−1^)	q_2_ (g/g)	R^2^
	1.512	0.3997	0.9729	7.047	0.4063	0.9871

**Table 2 foods-13-01634-t002:** Thermodynamic parameters for the adsorption of GLPWA on activated charcoal at different temperatures.

Temperature (°C)	Langmuir Model			Freundlich Model			∆H^0^ (kJ/mol)	∆S^0^ (J mol^−1^ K^−1^)	∆G^0^ (kJ/mol)
	q_max_ (g/g)	K_L_ (L/g)	R^2^	K_f_ (L/g)	n	R^2^			
25	0.4581	6.7322	0.9889	0.3374	0.2236	0.8623	−5.220	−3.553	−4.161
35	0.4679	5.8734	0.9847	0.3386	0.2415	0.9140			−4.126
45	0.4783	5.7461	0.9866	0.3460	0.2380	0.8987			−4.090
55	0.4865	5.5089	0.9856	0.3499	0.2494	0.9371			−4.054

**Table 3 foods-13-01634-t003:** Yield and total sugar content of fractions eluted with different percentages of ethanol.

Component	Water	10E	20E	30E	40E	50E
Yield (%)	19.47	13.80	17.51	15.46	8.49	9.80
Sugar content (%)	76.98	79.25	57.44	99.75	75.80	82.83

## Data Availability

The original contributions presented in the study are included in the article, further inquiries can be directed to the corresponding author.

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
