# Peer review of "Liquid-Phase Adsorption Behavior of β-D-Glucooligosaccharides When Using Activated Carbon for Separation, and the Antioxidant Stress Activity of Purified Fractions"

_foods, 2024, doi:10.3390/foods13111634_

Round 1
Reviewer 1 Report
Comments and Suggestions for Authors
Manuscript describes the separation of oligosaccharides by activated carbon and subsequent purification. The manuscript is well describes and I feel this is suitable for this journal. Some are not cleared, as mentioned below,
1) in introduction, line 80; I am not sure what is the theoretical understanding? You want to stress ie, please mention more.
2) HPAEC's condition is required more. column, elution solution.
3) 3.1 optimization is strong expression. You only changed the condition of adsorption. If this is scale up, some changes will be obtained. t that time, transport phenomena and intrinsic reaction should be considered more.
4) Figure 3 (c), at saturated domain, a little bit down. Please revise illustrated line.
5) 50E is better in Fig. F and G of Figure 5. And mentioned the effect of the sequence of monomer and oligomer structures on the activity of cell. primary sequence, or some configuration structure effect on the cell?
Reviewer 2 Report
Comments and Suggestions for Authors
See attachment

Comments on the Quality of English LanguageModerate revision is required
Reviewer 3 Report
Comments and Suggestions for Authors
The manuscript discusses the adsorption behaviour of glucooligosaccharides on activated carbon described by two tested thermodynamic isotherm models. The authors additionally explored the antioxidant activity of purified fractions. The topic of the work is interesting and suitable for publication in the journal. However, the authors should consider the inclusion of additional information in the methodology section so that the protocols and procedures can be understood and followed by the reader. In some sections of the manuscript, additional discussions on the reported results and observations are also required.
I have enclosed my comments to the authors in the attached PDF file.
